# Investigations Concerning the Influence of Sleep Disorders on Postural Stability in Young Men

**DOI:** 10.3390/ijerph19148809

**Published:** 2022-07-20

**Authors:** Anna Tetych, Grażyna Olchowik, Jan Warchoł

**Affiliations:** Department of Biophysics, Medical University of Lublin, K. Jaczewskiego 4, 20-090 Lublin, Poland; grazyna.olchowik@umlub.pl (G.O.); jan.warchol@umlub.pl (J.W.)

**Keywords:** posturography, sleep disorders, sleep quality, PSQI

## Abstract

Lack of sleep is a factor that disrupts the receptors’ reception of information from the environment and contributes to the emergence of problems with maintaining balance. The main aim of the study was to determine whether sleep disorders affect postural stability in young men. The study participants were 76 male students who were divided into groups with good and poor sleep quality. The division was made based on the results obtained from the questionnaire of the Pittsburgh Sleep Quality Index (PSQI). In each group, postural stability had been tested using three main tests: Sensory Organization Test (SOT); Motor Control Test (MCT); and Adaptation Test (ADT). The results of the analysis show that the obtained results differ in the examined groups under the SOT test. Different values of the tested parameters were noted among people with poor sleep quality and compared with the values of those who sleep well, which translates into a difference in the ability to maintain balance. The greatest impact is observed when using visual and a vestibular system to maintain a stable posture. It was confirmed that the lack of sleep significantly disturbs postural stability.

## 1. Introduction

Sleep is a basic human physiological need that regulates many important body functions. It influences hormonal balance, brain functions such as remembering, associating and concentration. Its disturbances significantly affect human wellbeing and intellectual performance during the day.

One of the sleep disorders is insomnia. It might be defined as subjectively perceived poor quality of sleep impairing functioning during the day. In young adults, insomnia is usually short-term (up to three months) due to the emergence of stress factors In this case the symptoms of insomnia disappear when the stress disappears or when the patient adapts to a new situation. Sleep disturbances may develop into long-term forms, lasting more than three months. The clinical form of insomnia includes problems with falling asleep, maintaining sleep continuity, and waking prematurely. A significant element of occurrence, prolongation and consolidation of sleep disorders are certain medications, past or current diseases and physical ailments. Irregular sleeping hours and improper preparation for sleep also seem important [1].

The consequence of disturbances in night sleep is daytime sleepiness and cognitive impairment [2]. The main negative effect is the deterioration of mental performance due to impairment of attention and logical thinking [3]. However, studies show that poor sleep also increases the risk of many other conditions, such as cardiometabolic diseases including hypertension [4,5], cardiovascular disease [6], obesity [7,8], diabetitis [7], stroke [9,10], depression [11], anxiety [12] and exerts a significant impact on disturbances in the proper functioning of various body systems.

The reduced concentration in people with poor sleep quality results in an increased need for alertness, which reduces stability [13]. There is also an impairment in the response to external stimuli [2]. It can also result in postural imbalances, which is observed especially in young people [14].

The balance control is defined as the complex interaction between the sensory input signals, the central integration of these input signals and the effective efficiency of movement [15]. Organs and mechanisms of the human body’s balance system report the position, speed and direction of movement of the body, and control the movement of the eyeballs to maintain a stable image of the environment and to generate a reaction of a posture that works against body oscillations from the balance position. The position of the Center of Gravity (COG) of the body is determined by information from atrial receptors, visual organs and somatosenoric systems. This information is analyzed in the central nervous system and used for the activation of the motion system to bring the body into equilibrium. The vertical posture is stabilized by minimizing the swaying motion. The vertical projection of the COG on the support plane shall not exceed the limits of the feet and the area between them [16]. Functional control of equilibrium allows coordinated motion in every environment. However, this requires the ability to react immediately to external change and to develop effective strategies for implementing body movement. Based on this, test requirements for the operational check for balance control or moving problems have been elaborated on [15]. One of these is the stability test.

The health assessment of the equilibrium control system might be carried out by means of multiple clinical tests [17]. However, not all of them are able to vary the patient’s performance in terms of quantity. Studies [18,19] indicate that it is reasonable to use Computerized Dynamic Posturography (CDP) for quantitative and qualitative assessment of the human balance system. The main parameter controlled during posturographic examinations is the position and speed of the projection of the center of gravity onto the base plane. CDP is a test of an individual’s ability to efficiently process individual inputs from the sensory system as to maintain balance control. This is achieved by interfering with the input signals of the sensory system in order to generate appropriate strategies for response to movement and posture. The usefulness of the sensory organs involved in postural control, the choice of appropriate motor strategy, and the latency of responses are assessed on the basis of the CDP study protocol [20].

An analysis of the literature shows that there are few studies that show a relationship between poor sleep quality and incorrect postural control in young people. Due to the specificity of the chosen area of education, students are a group of people in whom the only cause of imbalance control, excluding stimulants, seems to be poor sleep quality or its deprivation.

The aim of the study was to determine whether sleep disorders affect postural stability in young men. Moreover, the aim was also to assess the response of individual balance systems to poor sleep quality. Based on the literature review, two hypotheses were formulated:
Patients with poor and good sleep quality will have different results on the corresponding CDP tests;Poor sleep quality negatively affects the functioning of the balance systems.

## 2. Materials and Methods

### 2.1. Recruitment and Selection

A total of 101 men aged 20–23 years (Mean 20.58 ± 1.10 years) took part in the voluntary study. They were students of the Medical University of Lublin. The height of the subjects ranged from 170–196 cm (Mean 180.54 ± 6.15 cm). Based on a proprietary questionnaire, filled by all respondents, malformations, uncorrected visual defects, head and cervical spine injuries, arterial hypertension, bronchial asthma and chronic obstructive pulmonary disease, lipid and hormonal disorders and other chronic diseases were excluded in all subjects. None of the subjects had consumed alcohol in the week prior to the beginning of the study.

From the typed group of 101 people, based on a questionnaire they completed, 11 people were taking antihistamines, 3 were taking psychotropic drugs, 2 were taking strong painkillers, 1 was taking an antibiotic, 6 people reporting moderate dizziness and 1 complained of tinnitus. As such, they were excluded from further studies. The criterion for inclusion in the study was the resignation from caffeinated beverages on the day of the study. In addition, all participants gave their informed written consent to participate in the research in accordance with the procedures approved by the Ethics Committee of the Medical University of Lublin, approval number 0254/195/2011. Men had the option of resigning from participation in the study at any stage, without giving reasons.

Finally, 76 young men took part in the research. In the selected group, sleep quality was assessed based on the questionnaire PSQI (Pittsburgh Sleep Quality Index) [21]. The test consists of 18 test questions examining 7 elements of sleep quality: subjective sleep quality, time needed to fall asleep, sleep duration, sleep efficiency, sleep disturbances, taking hypnotic drugs and difficulties in functioning during the day. Each of the 7 elements is assessed on a scale of 0–3 points. A total score in the questionnaire of no more than 5 points means that the subjects’ sleep quality is correct. A score greater than 5 indicates poor sleep quality.

A selected group of men, on the day of sleep quality examination, were subjected to postural stability tests using the posturograph Equitest produced by NeuroCom International. Posturographic examination of Sensory Organization Test (SOT), Motor Control Test (MCT) and Adaptation Test (ADT) was performed.

### 2.2. Measurement Procedure

The basic test performed under the CDP is the SOT [20]. It assesses the patient’s ability to use sensory systems and adaptive responses of the central nervous system. The score for each test is computed by comparing the angular difference between the patient’s maximum anteroposterior COG displacement and the theoretical maximum COG displacement and is expressed as a percentage from 0 to 100. Results approaching 0 indicate rocking amplitudes approaching the stability limits, and a value of 100 indicates no anterior-posterior COG displacement of the body. The SOT examination was conducted in six conditions of stimulation of the sensory organs and the results are presented in Table 1.

As part of the SOT tests, the following analyses were carried out: Equilibrium Score (ES); sensory system; and the Movement Strategy (MS). All the tests provided for in the procedure were performed [22].

ES provides a quantitative assessment of the anteroposterior COG angular displacement of the body. ES values are calculated and presented as a percentage. The greater the displacement, the lower the ES value. ES1–6 constitute the arithmetic mean values of the results of the equilibrium analysis from three trials under test conditions SOT 1–6. The weighted average ES value from the six sensory stimulation conditions is the Composite Equilibrium Score (CES). This is an index of the overall balance of the body that accounts for the body’s COG displacement under all SOT conditions.

In order to check the usefulness of the signal from a given sensory system in the body balance control, the results of the sensory analysis are calculated and presented separately: for the somatosensory system (SOM = ES2/ES1); for the visual organ (VIS = ES4/ES1); for the vestibular organ (VEST = ES5/ES1); and for visual preference (PREF = (ES3 + ES6)/(ES2 + ES5)).

The result of the MS enables the quantification of the muscle activity responsible for the movements in the ankles and the muscle activity responsible for the movements in the hip joints during each attempt of the SOT test.

The MS is calculated based on the horizontal component analysis of force acting on the posturographic platform. It is compared with the maximum defined shear force value. This comparison is expressed as a percentage. If the calculated value in the test is 100, it means full use of the ankle strategy. The lower the test result, the greater the involvement of the hip joint in the balance strategy.

In order to investigate the short-term postural response, the latency was investigated as part of the MCT. During all tests, the tested individual remains in the same position, without bending the legs at the knees or leaving the platform. MCT is the test of the automatic postural response to a sharp change in the Center of Force (COF) position. The COF position measurements are used to calculate delay times in muscle response to an active force causing a sudden change in the COF position. The latency is defined as the time in milliseconds between the start of translation of the plate, which the subject is standing on, and the beginning of the postural response. The destabilizing movement of the strain gauge plates takes place in the horizontal plane under six conditions: low (LSB), medium (LMB) and large (LLB) retrograde translation; and analogously progressive (LSF, LMF, LLF). The range of displacements of these plates is selected with regard to the height of the examined individual. This is to give the COG’s the same angular velocity among individuals of different height. This elicits a summative postural response to back translation (LB) and forward translation (LF), which gets assessed in the test. On the basis of the determined values, the computer system calculates the total result of the postural response latency (LC) analysis.

The energy analysis during the long-term response to destabilizing stimuli was performed as part of the 5-step ADT. It is performed in conditions of destabilization of the platform, on which the tested individual is standing. This is achieved by unexpected rotation of the platform along the axis of the patient’s ankle joint, which produces a postural damping response to a disturbing somatosensory stimulus. Rotation moves the toes up (ADTU) or down (ADTD). The individual’s proper response is to return the body to an upright position. During five platform’s destabilization attempts (ADTU 1–5, ADTD 1–5), the central nervous system adaptively suppresses the response to the stretching impulses of the muscles stabilizing the body in the ankle. The ADT algorithms evaluate this adaptive process by determining whether the patient can reduce the amplitude and duration of the sway oscillation during successive destabilization attempts. The ability to suppress the swaying is determined by a dimensionless factor called sway energy. It takes values from 0 to 200 and the bigger the value, the bigger the COG deviation during the return to the equilibrium position.

The results of SOT, MCT and ADT were compared with the sleep quality test results using the Pittsburgh Scale [21].

### 2.3. Statistical Analysis

The results of the questionnaires together with the results of posturographic tests were saved and prepared for analysis in an MS Excel spreadsheet. Statistical analysis of the results was performed with the Statistica 13.3 software. The values of measurable descriptive traits of the studied men were based on the mean and standard deviation in the case of a normal distribution, and in the case of no normal distribution on the median and upper and lower quartile. Additionally, in both cases, the dispersion of the minimum and maximum values were examined. The significance level of all analyses was set at *p* < 0.05. In order to check whether the quantitative variable comes from a normally distributed population, the Shapiro–Wilk W-test was used. The homogeneity of variance in the studied groups was checked with the Levene’s test. In order to confirm the effect of poor sleep quality on the stability of the posture, the t-test was used, and if the parameters did not meet the assumptions of Levene’s test, the Mann–Whitney U test was used.

## 3. Results

Based on the results obtained from the PSQI questionnaire, participants were divided into two groups: control CG (Control Group) and experimental EG (Experimental Group). The first of them (CG) included 49 people who in the PSQI questionnaire received a score not greater than 5. The remaining 27 people received a score greater than 5 and were classified to the EG group. There were no significant differences in the demographic characteristics (Table 2) of both study groups in terms of age and height.

Table 3 presents the relationships between the results of the body balance analysis in the SOT test and the results from the PSQI of the respondents. The results of the comparisons of the mean and median values clearly show the influence of sleep disorders on the sensory response of the equilibrium system. Statistical significance was demonstrated in trials ES3, ES4, ES5, ES6, so each case of dynamic disturbances of the input signals of the sensory system. The results of the trials also determined the total CES balance score to be lower in people with poorer quality of sleep (Figure 1 and Figure 2).

Table 4 presents the relationships between the results of the sensory analysis and the results of the PSQI of the respondents. Based on the median, it was demonstrated that the SOM value confirming the usefulness of the signal from the somatosensory system did not change in the experimental group. The values of all other tests in the experimental group are lower than in the control group. However, statistical significance was only found for VIS and VEST (Figure 3).

Table 5 presents the relationships between the results of the MS and the analysis of sleep quality in the studied subjects. MS2 results do not differ in the compared groups, however, in the experimental group, the quartile range is greater, which means a greater spread of data. However, this comparison was not statistically significant. The values of all other parameters are clearly higher in people without sleep disorders. Statistical significance was demonstrated only when comparing the parameters MS3 and MS4. Furthermore, the quartile range has a greater value in the experimental group, and the graphs presented in Figure 4 confirm a greater dispersion of data in them, with an indication of lower values.

Table 6 shows the relationship between the results of the MCT and the analysis of the quality of sleep in the studied subjects. There was no significant difference in the results of the comparison of the latency time in people with and without the insomnia. The statistical analysis of mean and median values also did not provide evidence of a negative effect of poor quality of sleep on this type of posturographic examination.

Table 7 presents the relationships between the results of the ADT and the analysis of poor sleep quality in the studied subjects. A lack of statistical significance of the comparative analysis of results has been demonstrated in people with and without absence of insomnia. Comparing the mean and median values also does not provide evidence of a negative effect of poor quality of sleep on this type of posturographic analysis.

## 4. Discussion

The results obtained from the posturography tests indicate a number of significant differences between the studied groups of healthy individuals and those with sleep disorders. They show clearly that patients with diagnosed sleep disorders have problems with maintaining balance. This might be caused not only by dysfunction of the vestibular system, but also by the interaction of multiple impairments [23].

The cause of patients’ dysfunction of the information used in the vestibular system cannot be specified with one test only. The results of many tests, on the other hand, using the influence of disruptions of entering signals to the somatosensory and vision system can defy the functional deficit of the balance system in patients [2,19,24].

Current patterns assume that if an individual stands on a stable surface, 70% of stimulants come from the somatosensory system, about 20% come from the vestibular system and only 10% from the vision system [20]. On an unstable surface, up to 60% of the sensory information comes from the vestibular system, about 30% is visual and 10% comes from the somatosensory system. Depending on the mode of CDP testing, it is possible to determine sensory dysfunctions and also the inability of the postural control system to effectively use information from a given sensory system in order to maintain balance.

The results of the ES1 and ES2 (Table 3) performed with the platform stationary are not significantly different between the groups compared. Since under these conditions up to 70% of the stimuli controlling the balance system originate from deep sensory receptors [24], this may imply a slight influence of sleep disturbance on this system. The result of the ES3 measurement, also performed on a stationary platform but with visual stimuli disturbed by environmental movement, forces the person to use the proprioceptive system to maintain stability. In this case the values of this parameter are significantly lower in persons from the experimental group.

It is estimated that with a stable surface only about 20% of the signals controlling the balance system can come from the vestibular system [20]. However, this does not mean that poor sleep quality can have a significant effect on the function of deep sensibility receptors. This is not confirmed by the SOM values (Table 3). The results of MCT (Table 5) also do not confirm that. Reactions in response to surface translation are elicited mainly by proprioceptive stimuli [20]. Vestibular and visual signals cannot initiate these responses [25]. The latency times of these automatic responses are not abnormal in patients with vestibular and visual stimuli impaired due to insomnia.

In the group of people with insomnia, the values of ES5 and ES6 are significantly lower (Table 3). In both cases the patient is cut off from visual stimuli or receives disturbed information from the visual system. Moreover, the patient is standing on an unstable surface. This means that the individuals with insomnia were not able, as healthy subjects, to use signals from the somatosensory and vestibular systems to maintain a stable posture. Their responses were dependent on visual stimuli, even if they were imprecise. Thus, sleep disorders may result in inefficient use of cues from the vestibular system. Due to the significantly lower value of ES3 in people from group EG, it can be concluded that these people were excessively dependent on visual impulses [20]. The effect of sleep deprivation on visual perception in young people is similarly documented. Batuk et al. [26] showed that there was an increased risk of falls and loss of dexterity in balance control following sleep deprivation. The authors suggested that this is due to abnormalities in the processing of the visual signal, but not from disturbed atrial signals. This does not negate the relationship between dizziness and the impaired reception of vestibular signals. In order to assess this, however, it is necessary to take into account the results of various tests, not limited to posturography [27].

According to literature reports, chronic poor sleep quality also disturbs posture control in the absence of visual stimuli. The Furtado’s studies [13] of balance tests used to assess posture control were measured with the Biodex Balance System and compared between two groups with different sleep quality. The results showed that the lack of visual stimuli significantly worsened postural balance in people with chronic sleep insufficiency. The results were confirmed, despite the lack of significant differences in the results of the PSQI questionnaire, in the compared groups, but confirmed by other methods. Similar conclusions were drawn by Al-Rashid’s studies with the use of an isokinetic dynamometer [23]. The lack of visual stimuli reduces the stability indexes in people with sleep disorders, affecting the OSI Overall Stability Index as well as the MLSI lateral stability and the anterior-posterior APSI. Almojali’s studies, on the other hand, indicate that the exclusion of the visual system from posture control means increased activity in the vestibular system, the work of which may be impaired by worse sleep quality [28].

The clinical interpretation of the results of posturography [20] only excludes the isolated influence of poor sleep quality on the vestibular system. In our own research the combination of differences in the values of the ES3, ES5 and ES6 (Table 3) tests in the studied groups is partially confirmed by the results of the sensory analysis. The VEST value differs significantly in the compared groups, while the median value for PREF is lower in the experimental group, but the statistical analysis does not confirm this (Table 3). In this case only ES5 and ES6 parameters should be lowered in the tested experimental group, however the other comparisons should remain at a similar level [20]. In the tested cases, however, a significant difference and deterioration in the stability conditions in individuals with poor sleep quality were found, as well as being found in the test on unstable ground, with normal visual stimuli—ES4. This means that when somatosensory signals are inaccurate, patients require a stable surface to maintain proper balance. In the absence of a stable posture, sleep disorders impair the effective use of vestibular and visual stimuli [29]. This seems to be clearly confirmed by the lower values of VIS and VEST in persons with sleep disorders (Table 4).

With the exception of a measurement with closed eyes and a stable surface, the results of MS tests in individuals with poor sleep quality are lower than in the group of healthy ones. However, only the differences in the values of MS3 and MS4 are statistically significant. In both tests, signals from somatosensory and visual systems are interchangeably disturbed and used. This means that when these systems are disrupted, the individuals with sleep disorders have a greater preference for the hip strategy. This is confirmed by Fabbri in his research, who claims that patients with the correct balance strategy use mainly the ankle joint [30]. Balance disorders cause COG to approach the limits of its stability, which in turn increases the use of hip movements [20]. The same relationship can be found in the studies of other authors, e.g., when comparing the equilibrium strategies in people with a worse (weakened) condition of the equilibrium system. In Mei-Yun Liaw studies [31], the elderly individuals had a higher degree of postural imbalance and used the hip strategy to a greater extent to maintain balance, especially when standing on a swaying support surface in the absence of a visual environment or a disturbed visual environment. Older people required longer reaction times and showed less directional control in maintaining balance.

The change in movement strategy is not due to muscle fatigue resulting from functional impairment in persons with sleep disorders [32]. Despite the significant influence of height on the results of the balance studies [33,34,35], the length of the body is of no importance for the choice of the balance strategy. This allows us to conclude that sleep disturbances were the only predictor of changes in movement strategy of individuals in the authors’ own research.

The results of the ADT (Table 6) do not resolve the issue of the influence of sleep disorders on the ability of the patients to adapt to disturbing somatosensory stimuli caused by unexpected changes in the orientation of the support surface. The patients reactions related to movement planning [25,36], balance strategies and the adaptation process was not significantly dependent on the occurrence of insomnia. According to Tanwar’s study, balance is one of the main measures of muscle and neurological deficit [37]. Given that the somatosensory system also includes sensory perception, the central nervous system maintains adaptive abilities to suppress automatic responses to posture destabilization.

## 5. Conclusions

In conclusion, the study found that in the group of young men, poor sleep quality is a factor that significantly influences postural stability. It has been shown that CDP can be used to confirm this effect in a group of young men with sleep disorders. Men from the experimental group with confirmed poor sleep quality obtained worse results than those in the control group in the SOT tests: ES 3–6; CES; VIS; VEST; MS 3–4. The hip strategy is the preferred method of maintaining balance for individuals with poor sleep quality. Thus, the hypothesis number one was confirmed.

The influence of sleep disorders on the vestibular and visual systems was confirmed, but there was no effect on the somatosensory system. So, the hypothesis number 2 was partially confirmed.

In addition, according to the study, the education in proper sleep hygiene among students is necessary.

## Figures and Tables

**Figure 1 ijerph-19-08809-f001:**
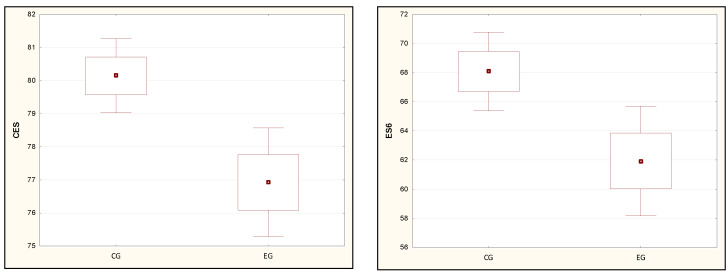
Results of studies of statistically significant parameters CES and ES6 meeting the conditions of normal distribution. Note: a dot—mean, a box—mean ± standard deviation, a whiskers—maximum and minimum values. CG: control group, EG: experimental group.

**Figure 2 ijerph-19-08809-f002:**
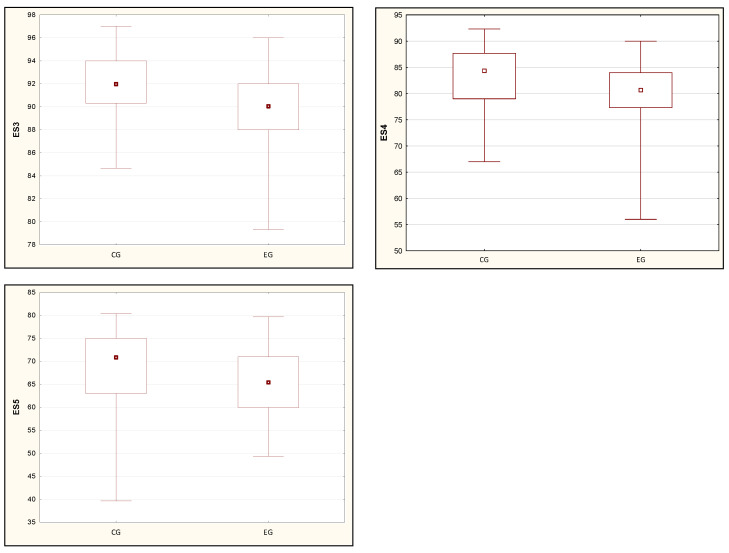
The study’s result of statistically significant differences in the values of ES3, ES4 and ES5 tests, which did not meet the condition of the normal distribution. Note: a dot—median, a box—interquartile range (IRQ = Q3 − Q1), a whiskers—maximum and minimum values. CG: control group, EG: experimental group.

**Figure 3 ijerph-19-08809-f003:**
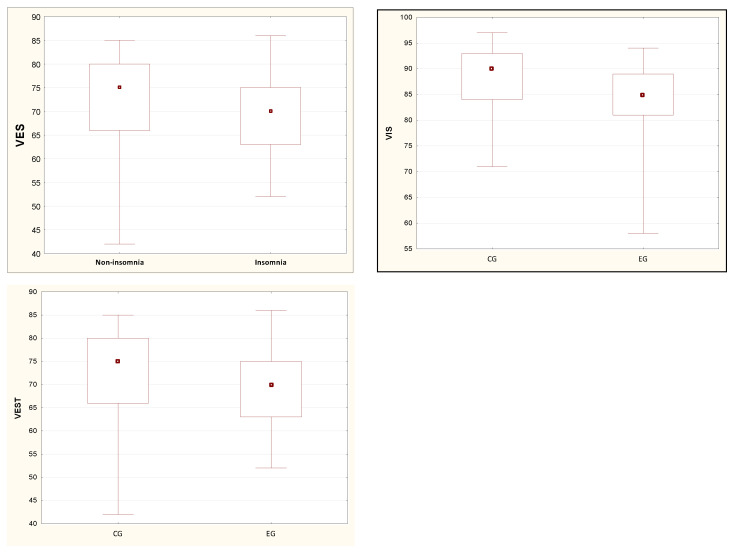
The results of study of statistically significant differences in the values of VIS and VES tests, which did not meet the condition of the normal distribution. Note: a dot—median, a box—interquartile range (IRQ = Q3 − Q1), a whiskers—maximum and minimum values. CG: control group, EG: experimental group.

**Figure 4 ijerph-19-08809-f004:**
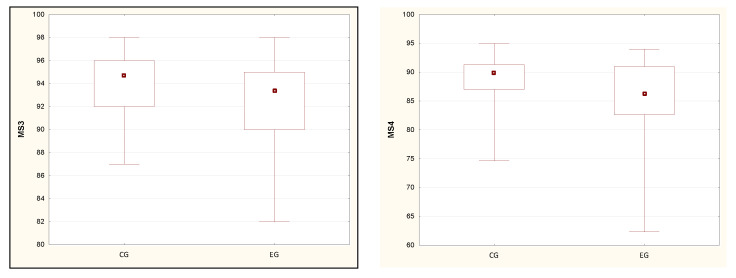
The results of the study of statistically significant differences in the values of MS3 and MS4 tests, which did not meet the condition of the normal distribution. Note: a dot—median, a box—interquartile range (IRQ = Q3 − Q1), a whiskers—maximum and minimum value.

**Table 1 ijerph-19-08809-t001:** Conditions of stimulated sensors of the sensory organization test [20].

Condition	Vision	Platform	Disturbance	Using
SOT1	Eyes open	Fixed		Somatosensation
SOT 2	Eyes closed	Fixed	Vision	Somatosensation
SOT 3	Eyes open, Swaying visual surround	Fixed	Vision	Somatosensation
SOT 4	Eyes open	Swaying	Somatosensation	Vision
SOT 5	Eyes closed	Swaying	Somatosensation, Vision	Vestibular
SOT 6	Eyes open, Swaying visual surround	Swaying	Somatosensation, Vision	Vestibular

**Table 2 ijerph-19-08809-t002:** Respondent’s demographic characteristic.

Demographic Characteristic	CG (49 Persons)		EG (27 Persons)		
	Mean	SD	Mean	SD	*p*
Height (cm)	180.41	3.64	179.78	3.46	0.46
Age (year)	20.51	1.04	20.63	0.93	0.62

Note: CG: control group, EG: experimental group.

**Table 3 ijerph-19-08809-t003:** Sensory Organization Test (SOT)—Equilibrium Score (ES) and Composite Equilibrium Score (CES) (ES scores correspond to individual SOT conditions).

	CG		EG		
Test	Mean (SD)	Med (IRQ)	Mean (SD)	Med (IRQ)	*p*
ES1 [%]	94.44 (2.42)	95.00 (2.00)	94.41 (1.69)	95.00 (3.00)	0.57
ES2 [%]	93.54 (2.38)	94.00 (2.00)	93.30 (2.92)	94.00 (4.67)	0.87
ES3 [%]	91.96 (3.00)	92.00 (3.67)	90.09 (3.82)	90.00 (4.00)	0.04 *
ES4 [%]	83.16 (6.01)	84.33 (8.67)	79.36 (7.61)	80.67 (6.67)	0.03 *
ES5 [%]	68.15 (9.09)	71.00 (12.00)	64.80 (6.99)	65.33 (11.00)	0.02 *
ES6 [%]	68.08 (9.64)	67.67 (14.00)	61.94 (9.96)	62.00 (12.33)	0.01 **
CES [%]	80.14 (3.99)	80.00 (6.00)	76.93 (4.35)	77.00 (8.00)	0.01 **

Note: * *p* < 0.05—statistically significant result (Mann–Whitney), ** *p* < 0.05—statistically significant result (*t*-test). CG: control group, EG: experimental group.

**Table 4 ijerph-19-08809-t004:** Sensory Organization Test (SOT)—sensory analysis for the somatosensory system (SOM), the visual organ (VIS), the vestibular organ (VEST), and pattern preference (PREF).

	CG		EG		
Test	Mean (SD)	Med (IRQ)	Mean (SD)	Med (IRQ)	*p*
SOM [%]	98.63 (1.83)	99.00 (2.00)	98.44 (2.52)	99.00 (2.00)	0.88
VIS [%]	88.18 (6.56)	90.00 (9.00)	84.15 (8.18)	85.00 (8.00)	0.03 *
VEST [%]	72.06 (9.31)	75.00 (14.00)	68.70 (7.53)	70.00 (12.00)	0.03 *
PREF [%]	96.71 (4.26)	99.00 (6.00)	95.30 (5.59)	97.00 (5.00)	0.17

Note: * *p* < 0.05—statistically significant result (Mann–Whitney). CG: control group, EG: experimental group.

**Table 5 ijerph-19-08809-t005:** Sensory Organization Test (SOT)—MS movement strategy corresponding to particular SOT conditions.

	CG		EG		
Test	Mean (SD)	Med (IRQ)	Mean (SD)	Med (IRQ)	*p*
MS1 [%]	95.17 (2.11)	96.00 (2.67)	94.89 (1.90)	95.00 (2.00)	0.31
MS2 [%]	94.48 (2.47)	95.00 (2.00)	94.57 (1.81)	95.00 (3.00)	0.84
MS3 [%]	94.04 (2.69)	94.67 (4.00)	92.02 (4.03)	93.33 (5.00)	0.02 *
MS4 [%]	88.68 (4.43)	90.00 (4.33)	85.38 (6.58)	86.33 (8.33)	0.01 *
MS5 [%]	77.80 (6.96)	80.00 (7.00)	74.72 (8.65)	77.00 (14.33)	0.15
MS6 [%]	81.69 (5.60)	84.00 (7.00)	79.57 (6.51)	81.00 (9.33)	0.16

Note: * *p* < 0.05—statistically significant result (Mann–Whitney). CG: control group, EG: experimental group.

**Table 6 ijerph-19-08809-t006:** Motor Control Test (MCT)—latency time for small, medium and large back translation (LSB, LMB, LLB) and forward (LSF, LMF, LLF). Average back translation (LB) and forward (LF). Cumulative result of postural response latency analysis (LC).

	CG		EG		
Test	Mean (SD)	Med (IRQ)	Mean (SD)	Med (IRQ)	*p*
LSB [ms]	142.45 (10.41)	140.00 (15.00)	141.11 (10.13)	140.00 (20.00)	0.58
LMB [ms]	135.61 (10.88)	135.00 (15.00)	133.33 (8.66)	135.00 (15.00)	0.25
LLB [ms]	129.49 (13.88)	130.00 (10.00)	128.52 (8.41)	125.00 (10.00)	0.61
LB [ms]	135.84 (8.50)	137.00 (7.00)	134.26 (7.13)	133.00 (14.00)	0.37
LSF [ms]	142.96 (10.39)	145.00 (10.00)	141.85 (12.18)	140.00 (20.00)	0.54
LMF [ms]	137.45 (8.95)	135.00 (15.00)	138.70 (11.98)	140.00 (20.00)	0.71
LLF [ms]	130.51 (10.13)	130.00 (10.00)	131.30 (8.61)	130.00 (5.00)	0.84
LF [ms]	136.98 (9.46)	137.00 (10.00)	137.26 (9.36)	135.00 (13.00)	0.90
LC [ms]	133.84 (9.56)	134.00 (10.00)	133.56 (8.08)	132.00 (14.00)	0.88

Note: CG: control group, EG: experimental group.

**Table 7 ijerph-19-08809-t007:** Adaptation Test (ADT)—sway energy for five upward (ADTU) and five downward (ADTD) ground destabilization.

	CG		EG		
Test	Mean (SD)	Med (IRQ)	Mean (SD)	Med (IRQ)	*p*
ADTU1	84.12 (20.22)	80.00 (25.00)	82.74 (18.83)	81.00 (32.00)	0.94
ADTU2	65.92 (15.63)	63.00 (20.00)	66.37 (14.26)	66.00 (17.00)	0.57
ADTU3	63.39 (13.88)	60.00 (18.00)	60.33 (12.74)	62.00 (23.00)	0.35
ADTU4	56.96 (10.39)	56.00 (15.00)	55.44 (12.40)	55.00 (17.00)	0.69
ADTU5	55.20 (8.95)	55.00 (12.00)	53.59 (10.95)	52.00 (14.00)	0.50
ADTD1	48.90 (10.13)	47.00 (13.00)	49.78 (9.86)	50.00 (15.00)	0.45
ADTD2	41.20 (8.50)	40.00 (13.00)	45.04 (10.96)	43.00 (16.00)	0.15
ADTD3	37.92 (9.46)	37.00 (10.00)	40.52 (12.56)	38.00 (14.00)	0.47
ADTD4	36.73 (9.56)	35.00 (10.00)	37.81 (9.31)	35.00 (10.00)	0.47
ADTD5	36.04 (8.25)	36.00 (9.00)	37.70 (8.43)	37.00 (13.00)	0.34

Note: ADTU—dorsiflexion of the feet, ADTD—plantar flexion of the feet. CG: control group, EG: experimental group.

## Data Availability

The data supporting reported results are available in the corresponding author.

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
