# Peer review of "Investigations Concerning the Influence of Sleep Disorders on Postural Stability in Young Men"

_ijerph, 2022, doi:10.3390/ijerph19148809_

Round 1

Reviewer 1 Report

The paper presents an appealing study on the potential influence of sleep disorders on postural stability in young men.

The study seems well conducted and the results are interesting both for research and practice.

The study sample is quite small. Please discuss potential limitations linked to the sample size.

Were groups comparable regarding potential confounders (age, sex, education, etc.)?

Questionnaires were used. This is another possible source of bias. Please discuss in more detail.

The conceptual relevance could be better highlighted.

Which conceptual models are advanced? How?

It is not clear if in this study sleep is seen as a predictor of posture or the other way round (or both are correlates of a third variable).

The practical implications could be made clearer with more detailed examples.

Authors speak about “lack of sleep”. This can be misleading (no sleep at all). Consider a different wording.

Please check English language (incl. in the title).

Reviewer 2 Report

Notes in the attachment.

Reviewer 3 Report

In general, the survey is well structured clearly and the results are fairly presented. Before publication, however, I would ask you to address a few concerns:

In the introduction, there is no information about the effect of sleep disorders on the risk of developing chronic conditions such as hypertension, etc. (https://doi.org/10.3390/jcm11082106)

In the methodology, the authors mention the health questionnaire used, but there is a complete lack of information about what the questionnaire was, who filled it out? What did it contain? One can guess from the text that it constituted the inclusion criteria. It certainly needs to be specified very carefully.

In the description of the scale, the authors cannot give how many patients obtained what score in the study, this is part of the results and should be there. Here, please provide only a description of the tools used.

Before using a particular abbreviation for the first time, please explain it.

Please correct the name of the test under the captions on the charts.
